# Identifying context-specific domains for assessing antimicrobial stewardship programmes in Asia: protocol for a scoping review

Huong Thi Lan Vu [ID],[1] Raph L Hamers,[2,3] Ralalicia Limato [ID],[2,3] Direk Limmathurotsakul [ID],[3,4] Abhilasha Karkey,[3,5] Elizabeth Dodds Ashley,[6] Deverick Anderson,[6] Payal K Patel,[7] Twisha S Patel,[7] Fernanda C Lessa,[7] H Rogier van Doorn[1,3]

**Correspondence to**
Dr Huong Thi Lan Vu;
huongvtl@oucru.org

## ABSTRACT

**Introduction** Antimicrobial stewardship (AMS) is an important strategy to control antimicrobial resistance. Resources are available to provide guidance for design and implementation of AMS programmes, however these may have limited applicability in resource-limited settings including those in Asia. This scoping review aims to identify context-specific domains and items for the development of a healthcare facility (HCF)-level tool to guide AMS implementation in Asia.

**Methods and analysis** This review is the first step in a larger project to assess AMS implementation, needs and gaps in Asia. We will employ a deductive qualitative approach to identify locally appropriate domains and items of AMS implementation guided by Nilsen and Bernhardsson's contextual dimensions. This process is also informed by discussions from a technical advisory group coordinated by the US Centers for Disease Control and Prevention to develop an AMS HCF-level assessment tool for low-income and middle-income countries. We will review English-language documents that discuss HCF-level implementation, including those describing frameworks, components/elements or recommendations for design, implementation or assessment globally and specific to Asia. We have performed the search in August–September 2021 including general electronic databases (MEDLINE, Embase, Web of Science and Google Scholar), region-specific databases, national action plans, grey literature sources and reference lists to identify eligible documents. Country-specific documents will be restricted to countries in three subregions: South Asia, East Asia and Southeast Asia. Codes and themes will be derived through a content analysis, classified following the predefined context dimensions and used for developing domains and items of the assessment tool.

**Ethics and dissemination** Results from this review will feed into our stepwise process for developing a context-specific HCF-level assessment tool for AMS programmes to assess the implementation status, identify intervention opportunities and monitor progress over time. The process will be done in consultation with local stakeholders, the end-users of the generated knowledge.

## STRENGTHS AND LIMITATIONS OF THIS STUDY

⇒ This scoping review will be guided by determinant frameworks that identify the context dimensions of implementation and by experts and current research to provide comprehensive and in-depth description of the relevant aspects of AMS programmes in general and in Asia.

⇒ We will employ a combination of search strategies that includes different types of databases to efficiently identify a broad spectrum of guidance documents from published and grey literature in English.

⇒ Local experts including academics and practitioners in AMS in Asia who are the end-users of this generated knowledge will be engaged during the review process.

⇒ We will not include original studies that reported results of AMS implementation from a limited number of single institutions or locations (due to the vast number of such reports).

## INTRODUCTION

The prevalence and incidence rates of antimicrobial-resistant (AMR) infections continue to rise, with higher levels of resistance reported from low-income and middle-income countries (LMICs) compared with high-income countries.[1] Misuse and overuse of antimicrobials in healthcare continues to be one major driver as consumption is accelerating worldwide especially in emerging economies. At the same time, access to life-saving drugs targeting the emerging resistant pathogens remains an issue in these settings while development of new drugs has been slow especially for the antibiotics against gram-negative pathogens.[2]

In this context, antimicrobial stewardship (AMS) is an important strategy in global and national action plans to control AMR. This has been defined in a number of ways

from technical descriptions focusing on prescriptions (eg, drug, dose, duration) to the concepts of responsibility in how antibiotics are used (eg, careful and responsible decisions to treat).[3] AMS has also been expanded beyond its core strategies and adult inpatient settings, with increasing roles of front-line providers (eg, nurses and pharmacists), rapidly evolving metrics for impact and success, and wide spectrum of implementation, engagement and practice.[4]

Guidance for AMS programmes are available from many resources such as those from WHO and the US Centres for Disease Control and Prevention (CDC),[5–7] expert reviews,[8 9] country and region-specific guidelines. The core elements and components of AMS programmes identified by several groups[10–13] often share similar aspects. These include senior management commitment, resource allocation, leadership, accountability, expertise on infection management and antimicrobials, specific actions to improve antimicrobial use, education and training, monitoring use and resistance epidemiology and reporting/feedback. These existing guidance and tools are usually not context specific, not tailored for resource-limited settings and not focused on implementation aspects of the programmes. The WHO toolkit for LMICs issued in 2019 and WHO policy guidance on integrated AMS activities issued in May 2021 discuss different sets of measures/indicators for assessing programme impact at the healthcare facility (HCF) level,[7 14] which include structural, process and outcome measures but these assessments are at a high level and do not include specific implementation contextual details of stewardship interventions in these settings. Therefore, existing guidance resources should be reviewed to develop a HCF-level tool that will guide implementation of AMS in hospitals in Asia.

## REVIEW RATIONALE

The existing guidance documents may have limited applicability to many countries in Asia where healthcare systems, the availability of diagnostic testing and antibiotics, public awareness and prescribing practices[8 15] and contextual and cultural characteristics[16 17] differ from the other regions. Therefore, contextualisation of the assessment of the status of AMS programme implementation is important. A 'glocalisation' approach has been discussed for reconciliation of global frameworks and methodologies and the local contexts in AMS implementation.[18] In November 2016, an expert panel consisting of 11 infectious diseases experts, researchers and opinion leaders from Asia developed a consensus statement on AMS programmes for acute care hospitals in the Asian region.[19] The panel identified common gaps and challenges facing AMS programmes, potential solutions and process and outcome measures for evaluating impact of the programmes. A checklist was designed based on existing materials. However, it provides limited in-depth understanding of the current state to support effective planning of targeted interventions. In May 2021, WHO issued a policy guidance on integrated AMS activities.[14] Within this guidance, WHO developed a list of essential national and HCF core elements to assist with strengthening AMS structures at both levels. Context-specific matters for HCF-level AMS implementation are not yet captured in the HCF tool. Understanding implementation needs and practices within the Asian context and being able to monitor progress over time are key to assist local healthcare facilities with improving antimicrobial use. A context-specific assessment tool that is practical and responsive to the local conditions is needed to support institutions to start and maintain AMS programmes in Asia. This assessment tool will be complementary to the WHO tools on national and HCF-level to provide support for hospitals in assessing their AMS status and improving the programme implementation.

## REVIEW OBJECTIVES

This scoping review protocol aims to describe our methods to identify the context-specific domains and items to develop a tailored assessment tool for Asia. Existing guidance documents will be analysed to extract these contextual domains and items related to an AMS programme in general and specific to Asian settings. This work constitutes the first step in a stepwise approach to the development of an integrated assessment tool that can be used to assess the current practices and needs and identify priorities for implementation and improvement of AMS programmes in Asia.

## METHODS AND ANALYSIS
### Conceptual model

To guide our data extraction and understanding of contextual features in AMS implementation, we will use the four levels of contextual dimensions (micro, meso, macro and multiple) as identified by Nilsen and Bernhardsson in their review of contextual determinants for implementation outcomes from 17 determinant frameworks.[20] We will employ a deductive qualitative approach to identify the domains and items of AMS implementation according to the four contextual dimensions as a framework for reference (table 1). This process is also informed by the discussions from a technical advisory group coordinated by the US CDC that has been formed to develop a HCF assessment tool focused on AMS implementation.

### Protocol design

This protocol follows the methodology for scoping review developed by Arksey and O'Malley[21] and Levac *et al*[22] through six steps:

### Stage 1: identifying the research question

This scoping review is designed to answer the following question: 'What are the domains and items that can constitute an assessment tool of AMS implementation in

**Table 1** List of contextual dimensions for programme implementation to guide the scoping review of AMS guidance documents

| Dimension | Description of initial themes and codes for data extraction |
|---|---|
| Micro level of healthcare | Patients' factors (knowledge, attitudes, preferences, needs and resources in antibiotic use and AMR) |
| Meso level of healthcare | Organisational culture and climate, readiness to change (commitment, preparation, prioritisation, efficacy and capacity to change, tension, practicality, flexibility), support (administration, staff, training resources, information and decision-support systems, expert support), structures (size, complexity, specialisation, differentiation, decentralisation) |
| Macro level of healthcare | Exogenous influences (policies, guidelines, research evidence, regulation and legislation, mandates, directives, recommendations, political stability, public reporting, benchmarking, organisational networks) |
| Multiple levels of healthcare | Social relations and support, financial resources (funding, reimbursement, incentives, costs), leadership (leaders, champions,etc), time restrictions, feedback (mechanisms for monitoring and providing feedback), physical conditions (equipment, facilities) |

AMR, antimicrobial resistant; AMS, antimicrobial stewardship.

Asia at a HCF level?' The assessment tool will aim to assess the state of AMS implementation in healthcare facilities and contextual factors that can influence its success and impact. It will also aim to identify challenges and opportunities, and help formulate priorities by generating actionable recommendations for next steps in the implementation of AMS programmes. Based on the conceptual model (table 1), we will identify the specific domains and items of four contextual dimensions that are important and relevant for AMS assessment and improvement at the

HCF in Asia. These domains and items will also be sent to a group of local experts in Asia for consultation on their relevance.

### Stage 2: Identifying relevant documents
We will search a number of electronic databases of published and unpublished literature to identify documents that provide guidance on AMS implementation globally and specific to Asia (table 2). Country-specific documents will be restricted to the countries within the

**Table 2** List of databases to be searched for the scoping review

| Database category | Databases |
|---|---|
| General database (a recommended optimal combination for searching systematic reviews)[24] | Embase<br>MEDLINE (including 'Epub Ahead of Print, In-Process, and Other Non-Indexed Citations')<br>Web of Science (Core Collection)<br>Google Scholar (first 200 references of relevance) |
| Regional database | China: Chinese biomedical literature Database (CBM)<br>India: IndMED<br>Korea: KoreaMed<br>South-east Asia: Index Medicus for the South-East Asia Region (IMSEAR)<br>China National Knowledge Infrastructure (CNKI)<br>Chinese Scientific Journal Database (VIP)<br>SaudiMedLit<br>Thai Index Medicus<br>Thai Journal Citation Index Centre<br>Thai Medical Index<br>Western Pacific Region Index Medicus (WPRIM)<br>WANFANG Data<br>Korea: RISS- Korean Education and Research Information Service<br>Asia: Bibliography of Asian Studies<br>Health Research and Development Information Network (HERDIN)<br>Japan: Cinii<br>Indonesia: Garba Rujukan Digital, Neliti |
| Grey literature database | ProQuest Dissertation and Theses Database (PQDT)<br>OpenGrey<br>Grey literature report |

three subregions of Asia: South Asia, East Asia and Southeast Asia. After this search, the study investigator group will review the search results to determine if documents identified have included the available AMS guidance resources commonly used and those that are mentioned to a lesser extent in the literature to make sure the search strategy has been appropriate and comprehensive. The search will be expanded to additional search engines available and manually checking all reference lists of included documents will also be performed to identify additional documents of relevance. We will also conduct a targeted search of the grey literature in local and international organisations' websites.

Following search terms will be used for the general databases (Embase MEDLINE, Web of Science, Google Scholar) as both keywords in the title and/or abstract and subject headings (eg, MeSH, EMTREE) as appropriate: ('antimicrobial stewardship' OR 'antibiotic stewardship') AND ('framework' OR 'guid*' OR 'tool*' OR 'recommend*' OR 'step' OR 'approach' OR 'policy'). For other databases, we will search using less restrictive terms ('antimicrobial stewardship' OR 'antibiotic stewardship') to be able to identify local relevant documents. Only documents published in English language will be included, with no restriction on years of publication.

### Stage 3: Selection of documents for review

This stage will be an iterative process: searching, refining the search strategy and reviewing articles for eligibility and inclusion into the final list of documents. The team will meet to discuss inclusion and exclusion criteria at the beginning of the process, and to review challenges and uncertainties and to refine the search strategy as needed along the process. The review process will essentially involve two steps. In the first step, two research assistants will independently screen the titles and abstracts of all citations returned from the search databases. Any documents considered relevant by either or both researchers will be included in the list for full-text screening at this step. In step two, the lead investigator (HTLV) together with the two research assistants will review the full-text of these documents and select those that fully meet inclusion/exclusion criteria for data extraction stage. Any discrepancies in the documents selected arising during the process will be discussed, and decision will be made through consensus within the team.

Documents will be included if they provide guidance for and/or input information that can help inform AMS implementation at the HCF level, including those describing frameworks, programme components and elements or recommendations for design, implementation or assessment. We will exclude any documents that only describe methods or results of research studies or reports on AMS programmes implemented at single healthcare facilities with a limited geographical scope. We will also exclude documents without identifiable author, publisher or year of publication. For guidance documents from the same institutions that were updated over time, we will only include the most updated version in the review.

### Stage 4: data extraction

Characteristics of included documents will be extracted using a data extraction form, which will be reviewed and updated along the extraction process as appropriate. Following information will be captured: confirmation on document's inclusion/exclusion criteria, publication year, geographical scope (global or regional), country, type of document (guidance, review, perspective/commentary, others), author, publisher and type (government agency, international body, expert group, individual authors, not-for-profit organisation, for-profit organisation, others). Data extraction will be based on the content published, no attempts will be made to contact authors for clarifications.

The content of the included documents will be read in-depth and coded following the context dimensions described by Nilsen and Bernhardsson. Initial themes and codes were developed based on the contextual dimensions as described in table 1. Codes are the specific summarising phrases; each code presents one specific area of AMS implementation that should be assessed. Themes are the higher-order categories consisting of multiple codes, each theme represents one broad area or construct of AMS implementation. The initial structure of themes and codes can be expanded during the extraction to add additional codes and themes and can be restructured if needed for clarity. We will use a deductive qualitative content analysis[23] approach that allows the analysis to evolve from our existing understanding of AMS implementation and the existing research about programme implementation. We will use NVivo software, a package for text-based analysis, to manage the texts and systematically organise our reading and coding. The qualitative content analysis is a popular, flexible and pragmatic method of analysing text data to attain a condensed and broad description of the topic through a systematic classification process involving coding and identifying themes and patterns.[23] The coding process will be conducted with close supervision by two experienced researchers of the research team. The research team will meet regularly to review the codes identified and ensure data extraction is consistent with the research question and purpose.

### Stage 5: Data summary and synthesis of results

After the documents were read and coded, we will summarise and organise the codes and themes into a cohesive and coherent structure. From this, we will identify possible items and domains that will be used to develop a HCF-level AMS assessment tool of AMS implementation. One item can be identified from the content of multiple codes and presented in a question format. One domain can be identified from the content of multiple themes that can help describe the items under it. Through this scoping review process, we will be able to achieve a broad and in-depth description of the various aspects of AMS

implementation that are applicable to Asian settings. This will help identify aspects commonly encountered in an AMS programme, but also point out the region-specific issues, challenges and opportunities and help the development of a practical and responsive assessment tool for use in these settings.

## Stage 6: Consultation

As this review aims to identify appropriate domains and items for the assessment tool specific to Asia, consultation with local experts on the identified content is an important step to check for validity and relevance. We will engage local experts who have experience with AMS in the region including academics and practitioners who will be the end-users of the knowledge generated from this scoping review. These local experts will be identified through our own individual networks in four countries where the investigator groups are working in (Indonesia, Nepal, Thailand and Vietnam) and from the published literature during our search. The identification and selection of the local experts are purposive in order to engage those with the most appropriate experience. The feedback from these local experts on the list of domains and items will be captured using a web-based survey. With the expected return rate of 30%, we plan to approach up to 90 potential experts (60 from the local experts in four countries and 30 from the literature.)

## Patient and public involvement

Patients or the public will not be involved in this scoping review. The assessment domains and items identified from this scoping review will be sent to local experts in AMS for their feedback on the relevance of these to be included in the assessment tool at HCF level.

## DISSEMINATION AND ETHICS

This scoping review is the first step in the overall project coordinated by the US CDC to assess AMS implementation in LMICs including Asia. The results from this scoping review will be used to develop a context-specific, assessment tool aiming at characterising the current state of AMS implementation and identifying interventions and opportunities for improvement. The scoping review process will only involve identification, selection and analysis of documents available for use in the public domains, therefore an ethics approval is not required in this study. The whole process of assessment tool development will be done in consultation with local stakeholders who are the end-users of the generated knowledge.

## Author affiliations
[1]Oxford University Clinical Research Unit, Ha Noi, Viet Nam
[2]Eijkman-Oxford Clinical Research Unit, Jakarta, Indonesia
[3]Centre for Tropical Medicine and Global Health, Nuffield Department of Medicine, University of Oxford, Oxford, UK
[4]Mahidol Oxford Tropical Medicine Research Unit, Bangkok, Thailand
[5]Oxford University Clinical Research Unit - Nepal, Kathmandu, Nepal
[6]Duke Antimicrobial Stewardship Outreach Network, Duke Center for Antimicrobial Stewardship and Infection Prevention, Duke University, Durham, North Carolina, USA
[7]Centers for Disease Control and Prevention, Atlanta, Georgia, USA

**Contributors** HTLV conceived of the idea, developed the research question and study methods. RLH, RL, DL, AK, EDA, DA, PKP, TSP, FCL and HRvD reviewed and aided in developing the research question and study methods. HRvD supervised the work throughout. All authors contributed meaningfully to the drafting and editing, and approved the final manuscript.

**Funding** This work is part of the project entitled 'Understanding variations in antimicrobial stewardship (AMS) programmes in hospital networks in Asia through a newly developed context-specific tool' funded by US Centers for Disease Control and Prevention (CDC) through the Broad Agency Announcement (BAA) 75D301-21-R-71738. HTLV was also supported by the National Institute for Health Research (NIHR) (using the UK's Official Development Assistance (ODA) Funding) and Wellcome (Grant Reference Number: 216367/Z/19/Z) under the NIHR-Wellcome Partnership for Global Health Research. The views expressed are those of the authors and not necessarily those of Wellcome, the NIHR or the Department of Health and Social Care.

**Competing interests** None declared.

**Patient and public involvement** Patients and/or the public were not involved in the design, or conduct, or reporting or dissemination plans of this research.

**Patient consent for publication** Not applicable.

**Provenance and peer review** Not commissioned; externally peer reviewed.

**ORCID iDs**
Huong Thi Lan Vu http://orcid.org/0000-0002-9579-5576
Ralalicia Limato http://orcid.org/0000-0002-5306-3254
Direk Limmathurotsakul http://orcid.org/0000-0001-7240-5320

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
