## [Reviewer comments · BMJ Open]

ARTICLE DETAILS

TITLE (PROVISIONAL)	Identifying context-specific domains for assessing antimicrobial stewardship programs in Asia: Protocol for a scoping review
AUTHORS	Vu, Huong; Hamers, Raph; Limato, Ralalicia; Limmathurotsakul, Direk; Karkey, Abhilasha; Dodds Ashley, Elizabeth; Anderson, D; Patel, Payal; Patel, Twisha; Lessa, Fernanda; van Doorn, H. Rogier

VERSION 1 – REVIEW

REVIEWER	Maeda, Masayuki Showa University, Infection Control Sciences
REVIEW RETURNED	24-Feb-2022

GENERAL COMMENTS	Thank you for the opportunity to review the manuscript, describing the protocol for a scoping review of antimicrobial stewardship (AMS) in Asia. This scoping review aims to develop guidance for AMS implementation in healthcare facilities in Asia. The focus of this study is an important topic in the Asia region. The overall text is well written, but the methods state what was slightly unclear and include several concerns. I do have a few comments that might improve this manuscript: Major comments 1) P3L14 Strengths and limitations of this review; "~ identify a broad spectrum of guidance documents from published and grey literature in English and four local Asian languages" P8L53-54; "Only documents published in English language will be included," It is unclear the searching strategy of regional databases in four local Asian languages for the scoping review. 2) P714-26 Stage 1 Stage 1 should be much better specific described and clarified what will be done. 3) P7LL36-40 It is unclear what "represent the available AMS guidance resources commonly used" is pointing out. This process will include serious bias. 4) P11 Stage 6 The searching strategy and review process is a scientific approach. In contrast, the process of stage 6 is narrative and slightly unscientific. Who are the local stakeholders in each Asia region? This process
---

	will include serious bias. The identification/selection process of local stakeholders should be clarified.
--	--

REVIEWER	Padigos , Junel Sunshine Coast Hospital and Health Service, Intensive Care Unit
REVIEW RETURNED	09-Apr-2022

GENERAL COMMENTS	Comments and Suggestions Abstract Line 8 – We can't really combat antimicrobial resistance anymore. I suggest to the authors to reword this to reflect that the aim of AMS is to minimise the emerging rates of AMR and to preserve efficacy of current antimicrobials, rather than "combatting" it. We can't combat something that is ever-evolving – what we can do is to minimise the rapidity of its rising rates. Abstract Line 11 – Will you identify which countries in Asia are resource-limited? How? Will you look at GDP or any other measures? What will be your basis? Abstract Line 14 – Can you include and specify this "Four local Asian languages" on your methodology, and what were your basis of including these four Asian languages only? There are a lot of countries in Asia and the authors need to establish the rationale of selecting these four languages only. Article Lines 18-19 – My understanding is, the basis for your approach of using deductive qualitative content analysis was to compare the results of the scoping review against "existing understanding of AMS implementation and the existing research about program implementation". Is this purely CDC-based? Can this be clarified and stated in the protocol if what constitutes this? Is this done already? Or yet to be done? Is this the one laid out in Table1? If yes, clarify this. If not, do you need separate synthesis of "existing research about program implementation" or is this purely based on what is recommended from the technical advisory group as coordinated by CDC? I suggest to the authors to clarify framework of reference that you will use to compare your results against in order to synthesise your findings deductively. Article Lines 40-48 – Can you consider revising and breaking this very long sentence? It also complicated with the frequent use of linking word "including" and use of connector "and". Very well-written protocol with methodology described well; however, approach at synthesis of evidence needs to be clarified. My comments are minor. This scoping review that the authors are to conduct is very timely and relevant, particularly in Asia, and hopefully aid to the future of "glocalization" approach to AMS. All the very best.
--

VERSION 1 – AUTHOR RESPONSE

Reviewer: 1

Dr. Masayuki Maeda, Showa University

Comments to the Author:

Thank you for the opportunity to review the manuscript, describing the protocol for a scoping review of antimicrobial stewardship (AMS) in Asia.

This scoping review aims to develop guidance for AMS implementation in healthcare facilities in Asia. The focus of this study is an important topic in the Asia region. The overall text is well written, but the methods state what was slightly unclear and include several concerns.

I do have a few comments that might improve this manuscript:

Major comments

1)

P3L14 Strengths and limitations of this review; "~ identify a broad spectrum of guidance documents from published and grey literature in English and four local Asian languages"

P8L53-54; "Only documents published in English language will be included,"

It is unclear the searching strategy of regional databases in four local Asian languages for the scoping review.

Response:

We would like to thank the reviewer for pointing this out. We initially proposed to include 4 local Asian languages of the four countries where our overall project is being conducted (Indonesia, Nepal, Thailand and Vietnam). We then decided to review the English-language publications identified from the listed sources only due to practical reasons. However, we did not update the information in the section "Strengths and limitation of this review" accordingly before the initial submission. We have removed this information from the section in the revised manuscript.

2) P714-26 Stage 1

Stage 1 should be much better specific described and clarified what will be done.

Response: We have added two sentences to this paragraph to describe how we identify the domains and items stated in the research question as below:

"Based on the conceptual model (Table 1), we will identify the specific domains and items of four contextual dimensions that are important and relevant for AMS assessment and improvement at the HCF in Asia. These domains and items will also be sent to a group of local experts in Asia for consultation on their relevance."

3) P7LL36-40

It is unclear what "represent the available AMS guidance resources commonly used" is pointing out. This process will include serious bias.

Response: In the following original sentence: "After this search, the study investigator group will review the search results to determine if documents identified represent the available AMS guidance resources commonly used", we meant to describe the steps involved to ensure the search strategy to be comprehensive. To make this clearer, we have revised the sentence as follows:

"After this search, the study investigator group will review the search results to determine if documents identified have included the available AMS guidance resources commonly used and those that are mentioned to a lesser extent in the literature to make sure the search strategy has been appropriate and comprehensive."

4) P11 Stage 6

The searching strategy and review process is a scientific approach. In contrast, the process of stage 6 is narrative and slightly unscientific.

Who are the local stakeholders in each Asia region? This process will include serious bias. The identification/selection process of local stakeholders should be clarified.

Response: We appreciate the comment from the reviewer on this point. We include this stage at the end of the process to make sure the identified domains and items from the literature valid and relevant to the context of Asia. We used a purposive non-probability sampling approach to identify the experts because we think that this is an effective way to identify the experts with the most relevant knowledge and experience on AMS implementation in Asia. We have revised and added more description to this stage of consultation as follows:

“As this review aims to identify appropriate domains and items for the assessment tool specific to Asia, consultation with local experts on the identified content is an important step to check for validity and relevance. We will engage local experts who have experience with AMS in the region including academics and practitioners who will be the end-users of the knowledge generated from this scoping review. These local experts will be identified through our own individual networks in four countries where the investigator group are working in (Indonesia, Nepal, Thailand and Vietnam) and from the published literature during our search. The identification and selection of the local experts is purposeful in order to engage those with the most appropriate experience. The feedback from these local experts on the list of domains and items will be captured using a web-based survey. With the expected return rate of 30%, we plan to approach up to 90 potential experts (60 from the local experts in four countries and 30 from the literature.)”

Reviewer: 2

Dr. June! Padigos , Sunshine Coast Hospital and Health Service

Comments to the Author:

Comments and Suggestions

Abstract Line 8 – We can't really combat antimicrobial resistance anymore. I suggest to the authors to reword this to reflect that the aim of AMS is to minimise the emerging rates of AMR and to preserve efficacy of current antimicrobials, rather than “combatting” it. We can't combat something that is ever-evolving – what we can do is to minimise the rapidity of its rising rates.

Response: We thank the reviewer for the comment. We have replaced “to combat antimicrobial resistance” by “to control antimicrobial resistance” in the abstract and introduction.

Abstract Line 11 – Will you identify which countries in Asia are resource-limited? How? Will you look at GDP or any other measures? What will be your basis?

Response: We thank the reviewer for the comment. We used the term “resource-limited” to refer to the settings that might not have sufficient resources to implement the basic elements of AMS programmes, for example those without available and accessible microbiology services. We did not use this term to refer to a country. We have removed this term to avoid confusion in the abstract. The scoping review will help generate the domains and items for an assessment tool specific to Asia, which is conducted in conjunction with the U.S. Centers for Disease Control and Prevention to develop an AMS HCF-level assessment tool for low- and middle-income countries.

Abstract Line 14 – Can you include and specify this “Four local Asian languages” on your methodology, and what were your basis of including these four Asian languages only? There are a lot of countries in Asia and the authors need to establish the rationale of selecting these four languages only.

Response: This comment shares similar concern with the comment made by Reviewer 1 and we also would like to thank the reviewer for pointing this out. We initially proposed to include English-language documents and documents published in 4 local Asian languages of the four countries where our

overall project is conducted (Indonesia, Nepal, Thailand and Vietnam). We then decided to review the English-language publications identified from the listed databases and sources only due to practical reasons. However, we did not update the information in the section “Strengths and limitation of this review” accordingly before the initial submission. We have removed this information from the section in the revised manuscript.

Article Lines 18-19 – My understanding is, the basis for your approach of using deductive qualitative content analysis was to compare the results of the scoping review against “existing understanding of AMS implementation and the existing research about program implementation”. Is this purely CDC-based? Can this be clarified and stated in the protocol if what constitutes this? Is this done already? Or yet to be done? Is this the one laid out in Table1? If yes, clarify this. If not, do you need separate synthesis of “existing research about program implementation” or is this purely based on what is recommended from the technical advisory group as coordinated by CDC? I suggest to the authors to clarify framework of reference that you will use to compare your results against in order to synthesise your findings deductively.

Response: The framework of reference is the four contextual dimensions presented in Table 1. We have added some clarifications to refer to Table 1 as the framework of reference for this deductive qualitative analysis. We also refer to this again in the section “Protocol design” – Stage 1 for clarity. This framework is used to guide our coding of the content of the included documents and generating the domains and items.

Article Lines 40-48 – Can you consider revising and breaking this very long sentence? It also complicated with the frequent use of linking word “including” and use of connector “and”.

Response: We have split the sentence and reduced the use of linking words as suggested.

Very well-written protocol with methodology described well; however, approach at synthesis of evidence needs to be clarified. My comments are minor. This scoping review that the authors are to conduct is very timely and relevant, particularly in Asia, and hopefully aid to the future of “glocalization” approach to AMS. All the very best.

Response: We highly appreciate the reviewer’s feedback overall on the manuscript and on the synthesis approach. We have clarified the framework of reference for guiding our deductive qualitative analysis and hope this have improved the clarity of the methodology.

VERSION 2 – REVIEW

REVIEWER	Padigos , Junel Sunshine Coast Hospital and Health Service, Intensive Care Unit
REVIEW RETURNED	14-Jul-2022
GENERAL COMMENTS	Thank you for responding to the review comments and for revising the protocol with more details and clarity. I reviewed the amendments based on previous recommendations and I am satisfied with the outcome, particularly with how the results are synthesised. This is an important piece of work that will hopefully provide more insights and understanding of effective approaches to burgeoning AMR rates, particularly in Asia.